# Effect of Redox Switch, Coupling, and Continuous Polarization on the Anti-Corrosion Properties of PEDOT Film in NaCl Solution

Victor Malachy Udowo [1,2] 🟢, Maocheng Yan [1,*], Fuchun Liu [1,3,*] and En-Hou Han [1,3]

[1] Institute of Metal Research, Chinese Academy of Sciences, Shenyang 110016, China; victorudowo@mail.ustc.edu.cn (V.M.U.)
[2] School of Materials Science and Engineering, University of Science and Technology of China, Hefei 230026, China
[3] Institute of Corrosion Science and Technology, Guangzhou 510530, China
\* Correspondence: yanmc@imr.ac.cn (M.Y.); fliu@imr.ac.cn (F.L.)

**Abstract:** Conjugated poly(3,4-ethylenedioxythiophene) (PEDOT) film was electrochemically synthesized on stainless steel (SS). Redox interactions between the PEDOT film and the SS substrate were examined in 3.5 wt.% NaCl aqueous solution with the aid of electrochemical and spectroscopic analyses. The results show that the PEDOT film exhibited a barrier effect and mediated the oxygen reduction reaction, thus hindering ion diffusion to the steel substrate. Localized electrochemical impedance spectroscopy (LEIS) of the scratched area on the polymer film shows that PEDOT healed the defect by coupling with redox reactions on the steel surface to prevent charge localization and concentration. The electroactivity of the polymer film declined when PEDOT was polarized at potentials $>-0.7$ V. Prolonged exposure of the PEDOT film to dissolved oxygen in NaCl solution resulted in the polymer's over-oxidation (degradation), evidenced by the formation of a carbonyl group in the spectroscopic result. The degradation of PEDOT was attributed to chain scissoring due to hydroxide ion attacks on the polymer chain.

**Keywords:** conducting polymer; coating; stainless steel; electrochemical interaction; degradation





## 1. Introduction

Owing to high conductivity and the ability to protect metals under a defective coating, conducting polymers (CPs) such as polyaniline (PANI), polypyrrole (PPy), and polythiophene (PTh) derivatives are still choice materials for use in corrosion prevention. They are considered environmentally friendly substitutes for the hazardous hexavalent chromium in conventional coatings [1,2]. CPs may exist in the semiconducting neutral form, where electrons are added to the polymer backbone and/or conducting oxidized states when electrons are removed [3–5]. It is postulated that the redox processes on the CP stabilize the steel potential in the passive regime and aid the formation of a protective oxide layer on the metal [4]. In the process, the CP charge lost to metal oxidation is replenished by oxygen reduction, thereby stabilizing the steel potential in the passive region to hinder further corrosion reactions.

CPs are shown to protect against metal corrosion via mechanisms such as the barrier effect, anodic passivation, and release of dopant ions [5–8]. Galvanic coupling is an important aspect of corrosion protection by CPs, and oxidants (such as dissolved oxygen) are vital to sustaining the effect on CPs [6–11]. Torresi et al. [5] found that the galvanic coupling between PANI and metals (Fe, Ni, Cu, and Zn) strongly depends on the redox states of the CP. Most recently, Yan et al. [6,7] reported the cathodic protection of Al alloy by undoped CPs. Reports show that almost all p-doped CPs are oxidant toward Fe and its alloys [5–8].

However, the prolonged exposure of CPs to oxygen may result in decreased efficiency and over-oxidation (degradation) of the polymer film. In fact, oxygen catalyzes the oxidation/doping of PANI and Ppy [6–8]. The degradation of PANI is found to release carcinogenic toxins while Ppy is unstable, especially in the conducting state [8]. Research on metal corrosion inhibition by CPs has since been achieved using polythiophene derivatives. Among PTh derivatives, poly(3,4-ethylenedioxythiophene) (PEDOT) is remarkably known for its mechanical flexibility, excellent conductivity, and environmental stability, making it suitable for use in antistatic coatings, electrode material in batteries, supercapacitors, and fuel cells [8–11]. Following decades of research, PEDOT has been found to inhibit the corrosion of steel and Al alloys [8–16].

Zhang et al. [12] and Gao et al. [13] reported the significant corrosion inhibition effects of PEDOT on SS and Al alloy in aqueous solutions. Su et al. [14] showed that reduced PEDOT could promote passivation of the metal–CP interface, thus suppressing further corrosion reactions. Zhu et al. [15] demonstrated the corrosion mitigation of SS in a chloride solution by PEDOT, which served as the electron exchange medium to passivize the metal and reduce oxygen availability on the surface. Although the use of PEDOT in the corrosion protection of active metals has been reported in the literature [12–16], the research scope has been limited, and the underlying mechanisms have not been thoroughly investigated. Furthermore, the effect of redox processes on the corrosion protection mechanism of the PEDOT film has not been completely investigated.

In this work, PEDOT was electrochemically synthesized on SS. The redox behavior of the SS covered by PEDOT film was examined in NaCl aqueous solution at different polarization levels via electrochemical and spectroscopic analyses. The corrosion protection mechanism of the PEDOT film is also discussed.

## 2. Materials and Methods

### 2.1. Material and Substrate Preparation

The substrate used in this work was 316L stainless steel (SS), which has the nominal composition given in Table 1 [17]. Samples of 316L SS cut to 10 mm × 10 mm × 5 mm sizes were embedded in epoxy resin, leaving just 1 cm$^2$ of exposed surface area for use in the electrochemical tests as the working electrode (WE). The work surface of the samples was polished with 400-, 600-, 800-, and 1000-grit SiC paper. All the chemicals were analytical grade and bought from Sigma Aldrich, China. The PEDOT film was synthesized from an aqueous mixture containing 30 mM EDOT, 0.1 M LiClO$_4$, and 30 mM sodium dodecyl sulfate (SDS) as the dopant anion and electron-transfer mediator.

**Table 1.** Chemical composition of 316L stainless steel (wt.%).

| Element | C | Mn | P | S | Cr | Ni | Mo | Si | N | Fe |
|---|---|---|---|---|---|---|---|---|---|---|
| Weight (%) | 0.03 | 1.62 | 0.04 | 0.01 | 18.1 | 11.9 | 2.40 | 0.28 | 0.10 | Bal. |

The PEDOT film was electrodeposited on 316L SS at a deposition current of 5 mA cm$^{-2}$ for 120 s and then left to dry naturally at room temperature. The choice of the current density was based on previous reports on the effects of polymerization potentials on the synthesis of PEDOT [16]. Electrodeposition was performed via a potentiostat (Reference 600+, Gamry Instruments, Warminster, PA, USA) in a three-electrode configuration comprising a platinum (Pt) mesh as the counter electrode (CE) and a saturated calomel electrode (SCE) as the reference electrode (RE). The SS covered by PEDOT film was the working electrode during the electrochemical tests. The chemical structures of the EDOT monomer and the neutral and oxidized forms of PEDOT are given in Figure 1.

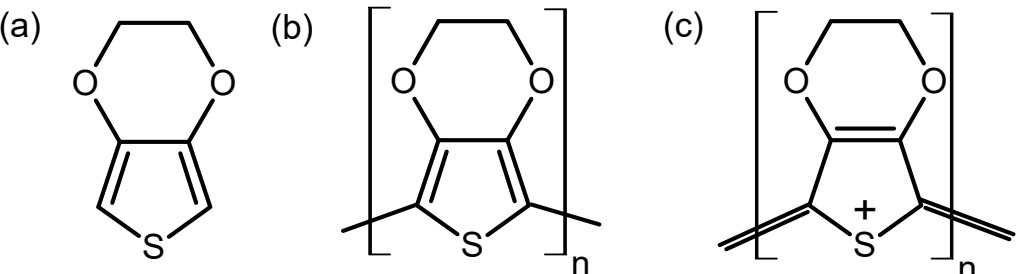

**Figure 1.** Chemical structures of EDOT monomer (**a**), neutral (**b**), and oxidized (**c**) forms of PEDOT.

*2.2. Electrochemical Measurements*

The electrochemical tests were performed in an aqueous solution of 3.5 wt.% NaCl as the electrolyte via the Gamry Reference/600+ potentiostat in a 3-electrode electrochemical cell configuration fabricated in a glass jar [18]. The cover of the glass cell had three openings for inserting the WEs (PEDOT-coated SS and bare SS) and a saturated calomel electrode (SCE) inserted in a Luggin capillary and Pt mesh as the CE [18]. Cyclic voltammetry (CV) was measured at a potential scan rate of 50 mV/s. The slow scan rate was chosen to allow sufficient time for the products of the reduction reaction in the forward scan to participate in the reverse reaction and generate clear anodic and cathodic reversible peaks. The electrochemical impedance spectroscopy (EIS) spectra were acquired at an open-circuit potential over the 110–0.1 Hz frequency range at 10 points per decade using an AC signal of 5 mV RMS amplitude.

Localized electrochemical impedance spectroscopy (LEIS) measurements were carried out using a VersaSCAN scanning electrochemical system (AMETEK, Berwyn, IL, USA) with low conductivity (approximately 70 μS/cm) to maximize resolution. A twin-electrode in a fixed position was placed above the defective coating surface measuring (1 cm$^2$) in NaCl aqueous solution. The microprobe tip travelled over the micro-defect on the coating surface at a separation distance of about 100 μm, stimulation frequency of 1 kHz, and disturbance amplitude of 10 mV. The generated LEIS data were converted to impedance using Origin software.

The PEDOT-coated electrode and bare SS were electrically coupled via the potentiostat in a zero-resistance ammeter (ZRA). In the second coupling experiment, the ability of oxygen to recharge (re-oxidize) the reduced PEDOT was examined. PEDOT was first reduced by coupling with bare SS, and the mixed potential and discharge current were recorded simultaneously. On decoupling, the individual electrode potentials were recorded to study the polymer reoxidation by DO. After a steady-state electrode potential was attained, the electrodes were coupled again, and the cycle was repeated three more times.

*2.3. Surface Characterization*

The morphology of the PEDOT coating after testing was observed using an optical microscope, scanning electron microscopy (SEM) with the aid of the SEM XL30-FEG spectrometer, and energy-dispersive spectrometer (EDS). After electrochemical tests, the PEDOT film was submitted to spectroscopic analysis with the aid of a (Jobin Yvon High resolution) Raman spectrometer equipped with a confocal microscope and a liquid-N$_2$ cooled charge-coupled device (CCD) detector and a He-Ne laser (632.8 nm). The chemical changes in the PEDOT film were further assessed by X-ray photoelectron spectroscopy (XPS, ESCALAB250). The details of the XPS analytical process can be found in the reference [19].

### 3. Results

#### 3.1. Electrodeposition of PEDOT on SS

The potential–time plot reveals the nucleation process of the PEDOT film on SS. Figure 2a presents two distinct regions. The electropolymerization process starts around 1.2 V, and at potentials higher than 1.6 V, the polymer passes into the oxidized state (Figure 2a). The SEM image of the freshly prepared PEDOT (Figure 2b) presents cauliflower morphology typical of electrochemically polymerized films [14–16]. The cauliflower-like islands show that the regions closer to the SS substrate have similar morphology (Figure 2b).

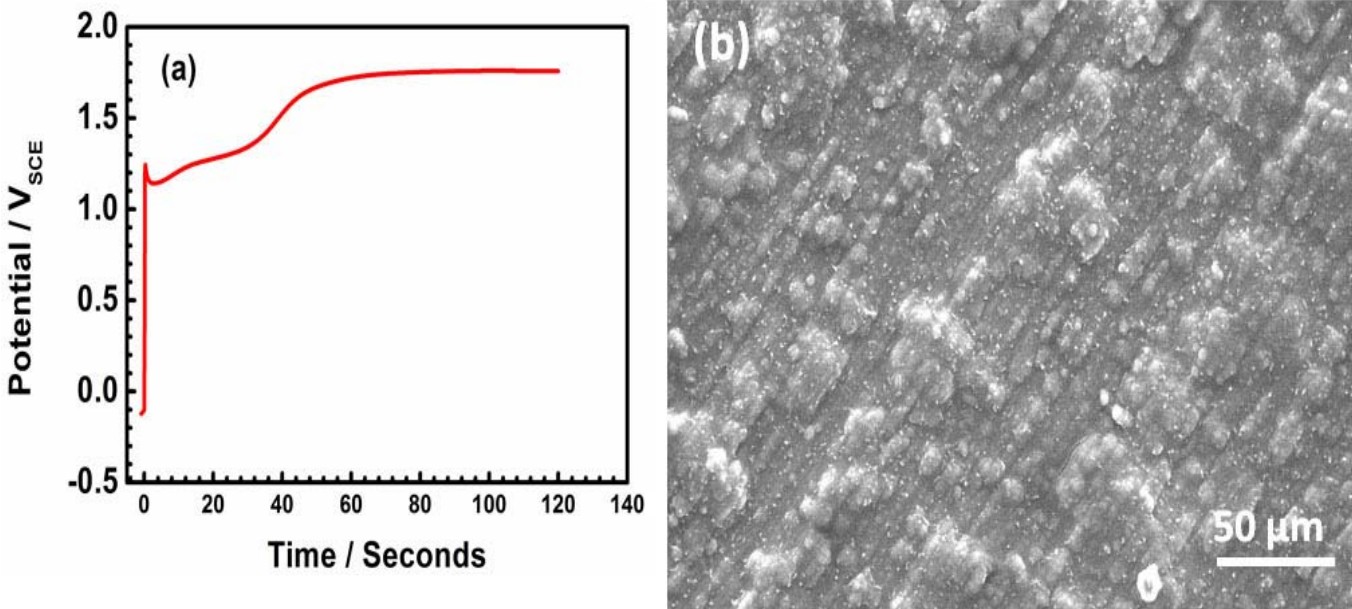

**Figure 2.** Potential–time plot recorded during electrodeposition of PEDOT on SS (**a**) and SEM image of freshly prepared PEDOT film (**b**).

#### 3.2. Cyclic Voltammetry

The CV of CPs in an electrolyte has been frequently used to evaluate their threshold reduction and oxidation potentials [20,21]. Figure 3 shows the CV curve of PEDOT film obtained in an aqueous solution of 3.5 wt.% NaCl. By visual inspection of the CV curve, the oxidation peak is observed around –0.2 V, while the reduction peak appears around –0.6 V. The peak separation and the decrease in the current density corresponding to the reduction peak (around −0.6 V) as the number of cycles increases indicates that the redox process on the polymer is quasi-reversible [21]. After 20 consecutive oxidation–reduction cycles, the CP's redox properties are almost unaltered. Such electrochemical stability makes PEDOT favorable for application in anti-corrosion coatings.

#### 3.3. Open-Circuit Potential

Figure 4 shows the open-circuit potential ($E_{OCP}$) of PEDOT and bare SS recorded as a function of the immersion period in NaCl solution. Protection of the PEDOT coating on 316L SS is evident (Figure 4) by the stabilization of the potential at about −0.0019 V after 43 h of testing, while that of bare SS evolves to correspond with iron dissolution. Whereas the early negative shift in the $E_{OCP}$ of PEDOT suggests a partial reduction of the CP by coupling with the SS substrate, reoxidation of the CP by oxygen may be involved in the later ennoblement [8–10]. Moreover, the potential difference between PEDOT and bare SS electrodes strongly suggests the existence of a galvanic effect, where the former is likely the anodic member.

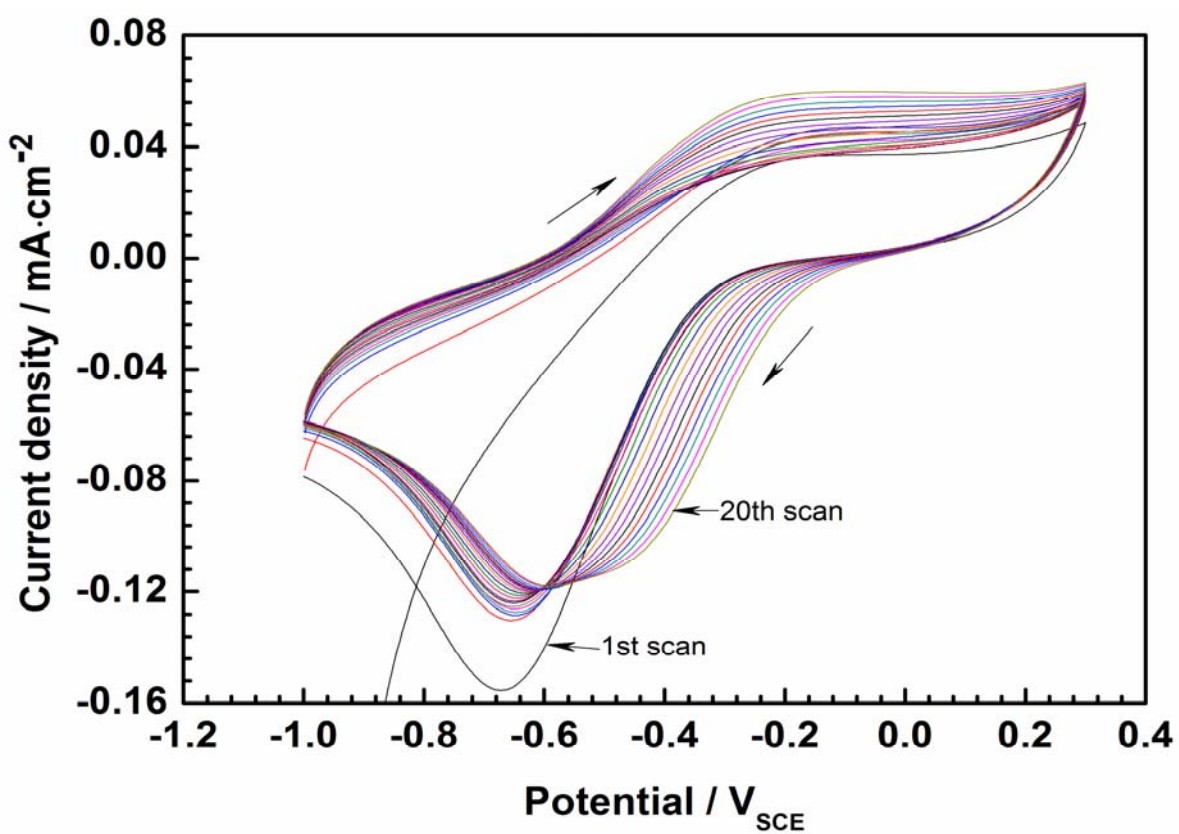

**Figure 3.** Cyclic voltammetry curves of PEDOT-coated electrode in 3.5% NaCl solution after 1 h of immersion to stabilize the electrode potential (scan rate is 50 mV/s).

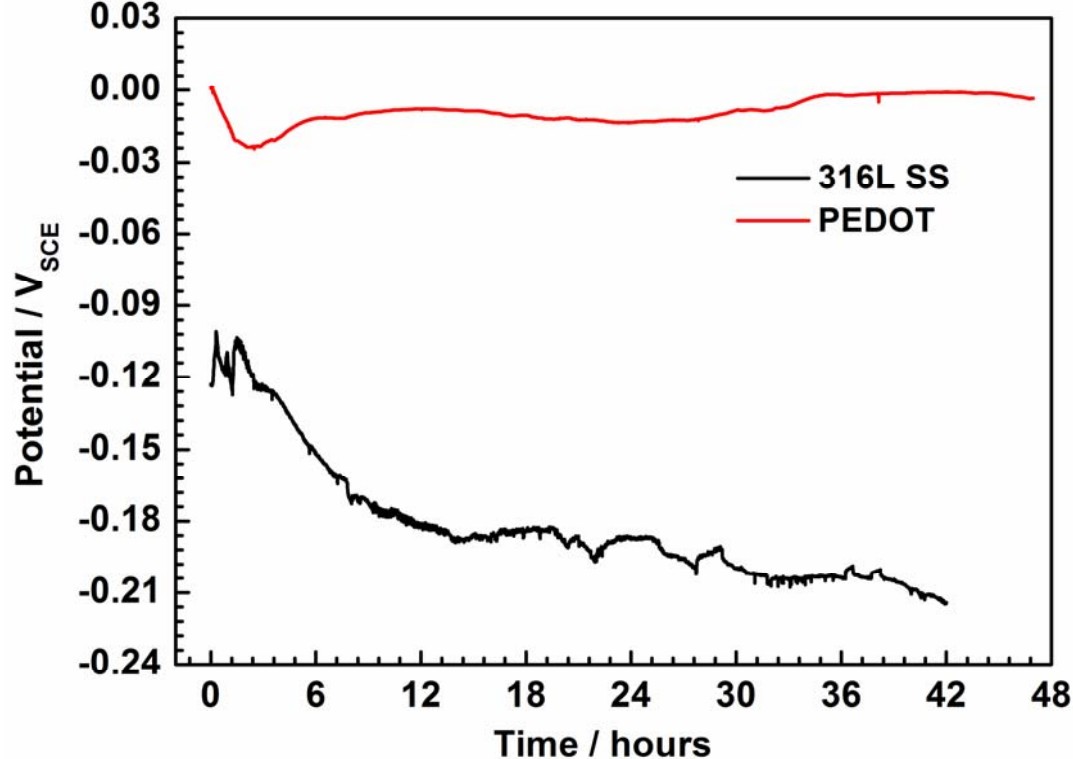

**Figure 4.** Evolution of open-circuit potential ($E_{OCP}$) of PEDOT-coated electrode and bare SS recorded as a function of immersion time in 3.5% NaCl solution.

### 3.4. Galvanic Coupling

The PEDOT-coated electrode (WE) and the bare SS (CE) were electrically coupled in NaCl solution via the potentiostat in ZRA mode, resulting in a steady coupling current of about 0.012 μA (Figure 5). The mixed potential (*E*mix) of PEDOT decreases from −0.060 to −0.168 V in 6 h (Figure 5), demonstrating the reduction of the polymer to its semi-conducting form. The extent of reduction is determined by the potential drift of the bare SS exposed to corrosion reactions in NaCl solution. After 6 h, an inflection is observed, and then the *E*mix ennobles significantly (Figure 5), suggesting reoxidation of the CP by coupling with the oxygen reduction reaction on the polymer film. Fluctuations in the mixed potential of PEDOT during coupling (Figures 5 and 6) suggest ionic exchanges occurring between the CP film and the electrolyte.

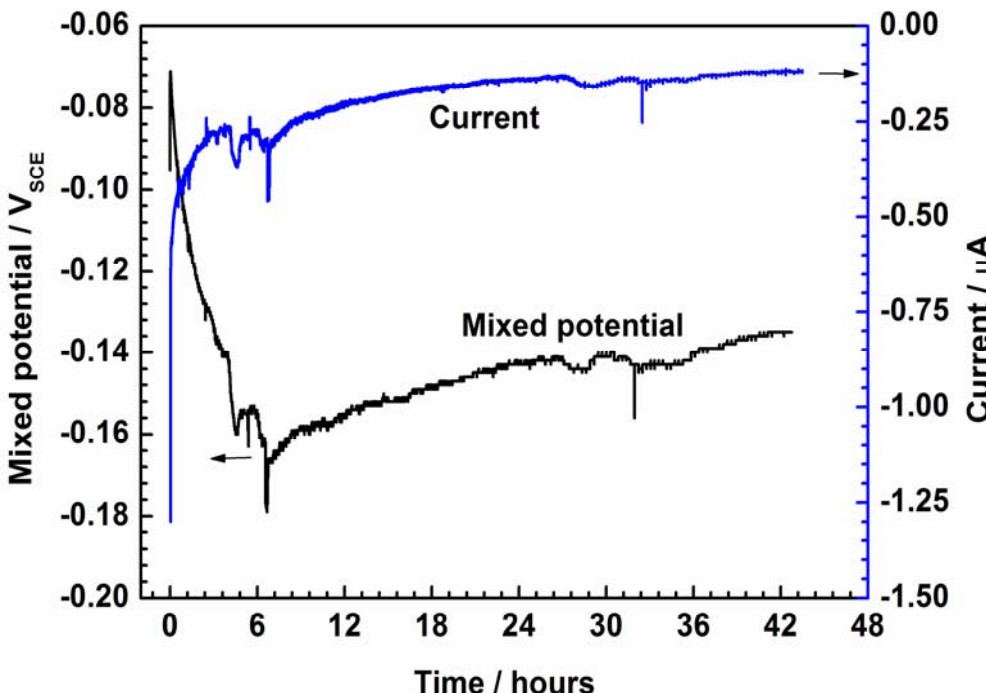

**Figure 5.** Galvanic coupling between the PEDOT and bare SS electrodes immersed in 3.5% NaCl solution.

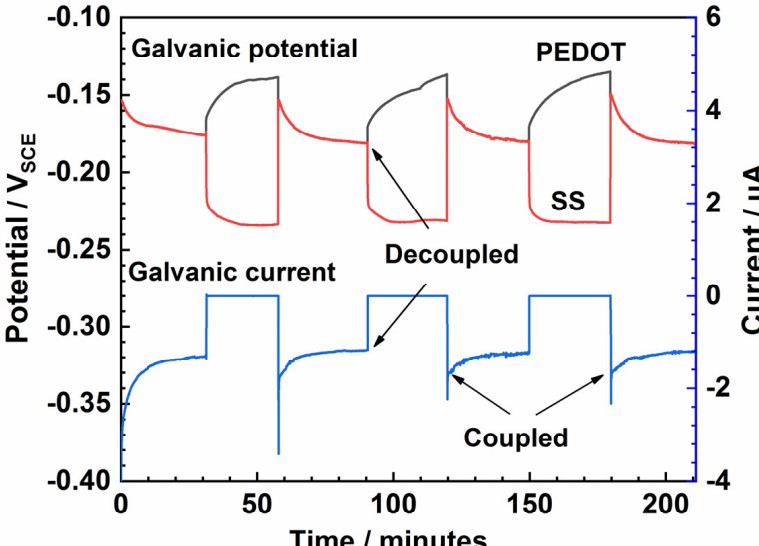

**Figure 6.** Dynamic galvanic corrosion in 3.5% NaCl solution, where PEDOT serves as the working electrode and SS is the counter electrode.

The capacity of DO to reoxidize the polymer film was examined with the aid of the classical charging/discharging experiments. First, PEDOT was reduced by coupling with bare SS for 30 min in NaCl solution and then reoxidized/recharged by decoupling while the $E$OCP of both electrodes was recorded concurrently. As seen in Figure 6, a discharge current of about 0.0736 μA is observed after 30 min. After four discharging/charging cycles, no noticeable change in the coupling current can be observed (Figure 6). However, the potential of PEDOT exhibits significant positive shifts on decoupling, while that of the bare SS shifts negatively, indicating that the electrodes exist in different oxidation states.

When two electronic conductors (such as the bare SS and PEDOT) are electrically connected, a purely electronic (i.e., non-redox) interaction is anticipated whereby the Fermi energy (or electrochemical potential of electrons) of each conductor is altered. As a result, electrons move from the (more active) SS into the PEDOT-coated electrode until the Fermi energies in the two phases are equal [8,9]. The immediate ennoblement in the potential of the PEDOT electrode upon decoupling reflects the Fermi level shift, owing to redox reactions on the polymer, which strongly buffer its electron activity (or Fermi energy).

The electrochemical behavior of the coated SS during the coupling experiments (Figures 5 and 6) is similar to that of the uncoupled PEDOT observed in Figure 4, which confirms that the initial potential decline is due to the coupling of redox reactions between the CP and the steel substrate in NaCl solution. However, the $E$OCP of the uncoupled PEDOT is more positive (Figure 4) because redox reactions on the coated substrate weakly discharge the charge stored in the CP film during electrodeposition. The $E_{OCP}$ of PEDOT is further discussed in Section 4.1.

*3.5. EIS*

The healing effect of the CP was studied using the LEIS maps of a simulated defect on the PEDOT film. The LEIS map of the defective coating displays a funnel shape after 0 h of testing in the NaCl solution (Figure 7a). In the early corrosion phase, the impedance at the defect is the minimum of the plot, about 768 Ω. Plateaus are observed around the defect in the areas covered by the coating, and the impedance is one order of magnitude higher than that of the defective area.

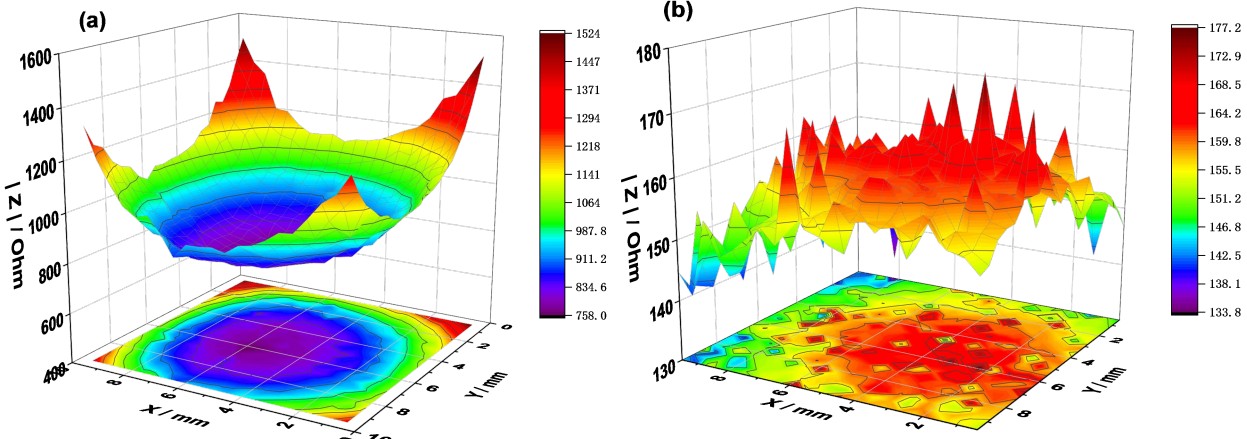

**Figure 7.** Localized EIS maps around the defect on PEDOT-coated SS after 1 h (**a**) and 6 h (**b**) of immersion in 3.5% NaCl solution.

After 6 h of immersion, the plateaus disappear due to redox interactions between the conductive polymer film and the steel substrate in NaCl solution. As expected, the impedance of the coated surface declines by an order of magnitude (Figure 7b). The uniformity in the impedance map of the defective coating after 6 h demonstrates the healing of the polymer film, which is partly due to the release of SDS anions when the polymer is reduced by coupling with the steel substrate. The formation of oxide corrosion products at the defect may also account for the effect observed in Figure 7b. Most importantly, the

PEDOT film prevents surface charge localization and concentration at the defect owing to its ability to conduct electrons and ions, resulting in the electrochemical protection of SS (Figure 7b).

The PEDOT film was subjected to conventional EIS tests in NaCl solution to examine the average impedance response of the entire surface. Figure 8 shows that the impedance of PEDOT decreases after 24 h of testing (Figure 8a,b). Such an evolution of impedance of CP-coated metals may not be entirely due to the barrier effect of the polymer film [22–24]. The galvanic coupling results show that PEDOT initially couples with the steel substrate (Figures 4 and 5), which will likely promote substrate oxidation, particularly in the early corrosion phase. The evolved cations will couple with the oxygen reduction reaction (ORR) in the NaCl solution to form a protective oxide layer, which hinders ion diffusion at the CP/steel interface [22–24].

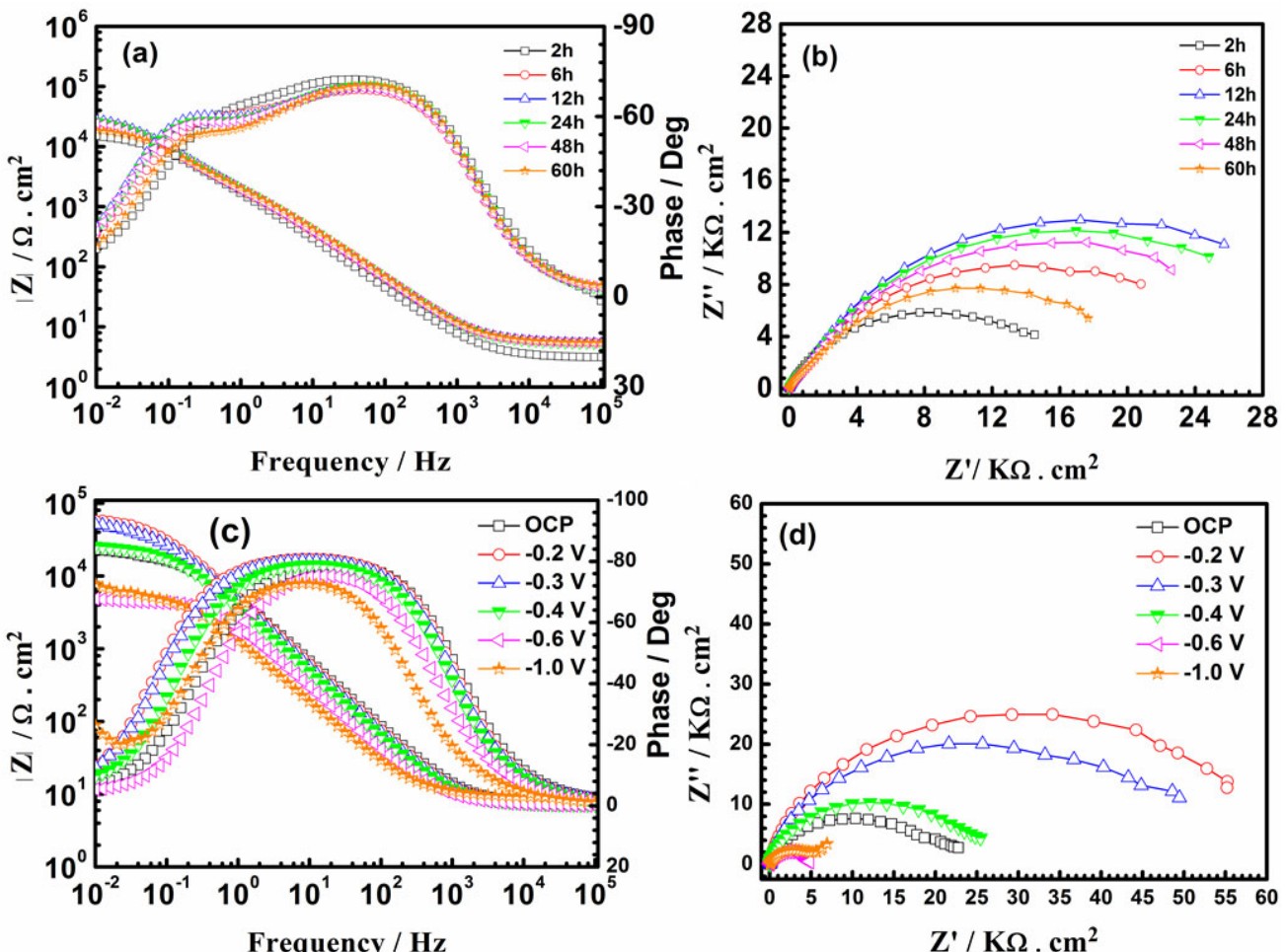

**Figure 8.** Bode and Nyquist plots of PEDOT-coated SS at open-circuit potential (OCP) (**a,b**), and the sample polarized at different potentials (**c,d**) in 3.5% NaCl solution.

The EIS data were further analyzed by fitting with the equivalent electric circuit in Figure 9, and the fitting parameters are given in Table 2. $R_s$ is the electrolyte resistance, and $R_{ct}$ represents the charge transfer resistance. The $R_{ct}$ arises from a kinetically controlled electrochemical reaction, such as electron transfer between the substrate and the CP [25,26]. The constant-phase element (CPE), denoted as $Q_{dl}$, replaces the capacitance due to the non-ideal capacitive response of the metal/coating interface. The CPE describes the heterogeneity, porosity, and roughness of the coating [25], which can be evaluated by the following equation:

$$CPE = Y_0 \, (\omega_m)^{n-1} \tag{1}$$

where $Y_0$ is the admittance of the CPE and $\omega_m$ is the frequency where the imaginary component of the impedance ($Z''$) has a maximum. The dispersion index (n) describes the system's deviation from a capacitor's ideal behavior. The value of n is such that $0 \leq n \leq -1$. Smaller n values imply more dispersion, while the effect ceases when n equals unity. Hence, the CPE behaves as an ideal capacitance [26]. The decreased $R_{ct}$ values of PEDOT after 24 h of testing in NaCl solution (Table 2) indicate penetration of the electrolyte through to the steel surface after the protectiveness of the oxide layer declines.

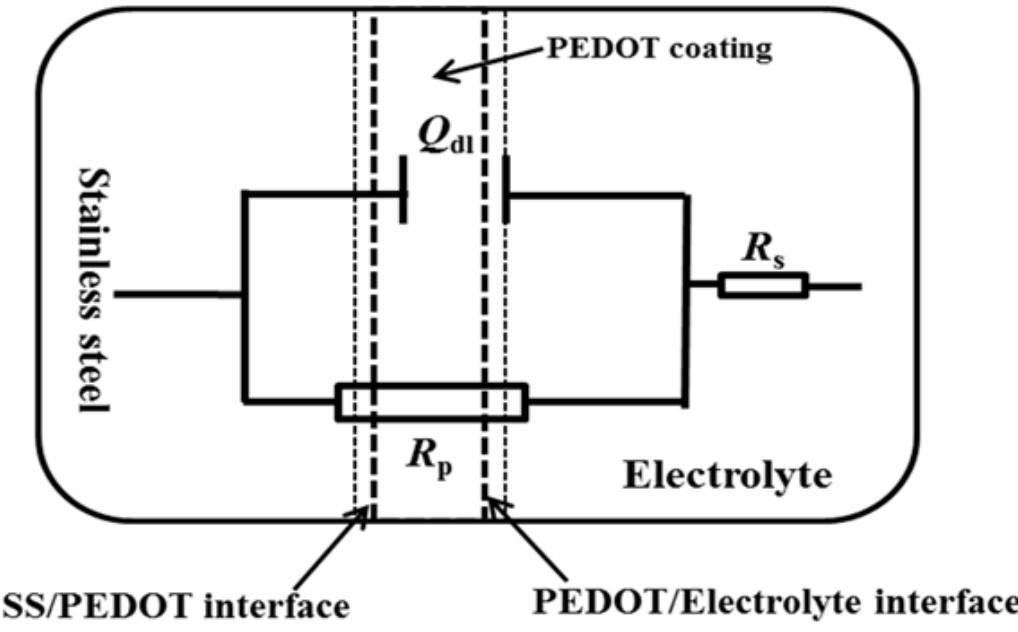

**Figure 9.** Equivalent electrical circuit used for fitting the EIS data of PEDOT-coated SS electrode using the ZSimpWin software. The chi square of the fit was $5.83 \times 10^{-4}$.

**Table 2.** Fitted electrochemical parameters of PEDOT from corrosion immersion test in 3.5% NaCl solution.

| Time/ Hours | $R_s$ $\Omega \cdot cm^2$ | $Q_{dl}$ $Y_0$ $Ss^n cm^{-2}$ | $n$ | $R_{ct}$ $K\Omega \cdot cm^2$ |
|---|---|---|---|---|
| 2 | 3.139 | $9.974 \times 10^{-5}$ | 0.8394 | 14.63 |
| 6 | 5.374 | $7.661 \times 10^{-5}$ | 0.8312 | 22.29 |
| 12 | 5.425 | $7.177 \times 10^{-5}$ | 0.8330 | 26.09 |
| 24 | 5.334 | $7.738 \times 10^{-5}$ | 0.8283 | 28.34 |
| 48 | 5.383 | $7.577 \times 10^{-5}$ | 0.8402 | 19.35 |

The redox behavior of PEDOT at various applied potentials was studied in NaCl solution. The PEDOT-coated SS was polarized at each potential for 300 s in NaCl solution before acquiring the impedance spectrum, starting from the OCP (0 V) to higher cathodic potentials (Table 3). At lower cathodic potentials (from $-0.2$ V to $-0.3$ V), the magnitude of $R_{ct}$ is double that of the undoped PEDOT (0 V). This may be related to the permeation of the electrolyte through the PEDOT film to the steel surface. In this potential range, the reduction of PEDOT and DO in the NaCl solution will occur, whereby $OH^-$ ions generated by the oxygen reduction reaction (ORR) can attack the PEDOT chains [27,28], resulting in the loss of electroactivity and conductivity.

**Table 3.** Fitted electrochemical parameters of the PEDOT polarized at different potentials in 3.5 wt.% NaCl.

| Polarized Potential $V_{SCE}$ | $R_s$ $\Omega \cdot cm^2$ | $Q_{dl}$ $Y_0$ $Ss^n cm^{-2}$ | $n$ | $R_{ct}$ $K\Omega \cdot cm^2$ |
|---|---|---|---|---|
| OCP | 6.618 | $4.045 \times 10^{-5}$ | 0.9098 | 22.92 |
| $-0.200$ | 7.413 | $3.847 \times 10^{-5}$ | 0.9071 | 56.63 |
| $-0.300$ | 7.380 | $4.308 \times 10^{-5}$ | 0.9010 | 50.66 |
| $-0.400$ | 7.354 | $5.071 \times 10^{-5}$ | 0.8929 | 25.81 |
| $-0.600$ | 7.212 | $8.780 \times 10^{-5}$ | 0.8757 | 4.836 |
| $-0.700$ | 6.947 | $7.194 \times 10^{-5}$ | 0.8742 | 2.393 |
| $-0.800$ | 7.152 | $4.775 \times 10^{-4}$ | 0.8224 | 4.018 |
| $-1.000$ | 8.844 | $2.756 \times 10^{-4}$ | 0.8438 | 7.728 |

The $R_{ct}$ of PEDOT exhibits a pronounced decrease after PEDOT was polarized from –0.4 V to –0.7 V, indicating a progressive reduction and loss of counterions from the polymer film (Table 3). The $R_{ct}$ gradually increases at potentials >–0.7 V, similar to the position of the cathodic peak in Figure 2, indicating a decline in the electroactivity and conductivity of the coated electrode. The diminishing electroactivity and conductivity at potentials >−0.7 V correspond to the abundance of OH⁻ ions generated from the excessive reduction of dissolved oxygen in this potential regime, which can attack the PEDOT chains and retard the electroactivity [27–30]. Fortunately, most electroactivity on CP can be preserved within the potential range from −0.4 V to −0.7 V, where the PEDOT electrode exhibits good conductivity and stability.

*3.6. Raman Spectroscopy*

The molecular changes in the polarized PEDOT film at different applied potentials were examined by Raman spectroscopy to clarify the observations in the EIS data. Sharp and well-defined bands characterize the spectra of polarized PEDOT (Figure 10). The most intense band at 1432 cm⁻¹ is assigned to the $C_\alpha = C_\beta$ symmetric stretch, and that at 1512 cm⁻¹ is the $C_\alpha = C_\beta$ asymmetric stretch [31–35]. The bands at 1364 and 1269 cm⁻¹ correspond to the $C_\beta = C_\beta$ symmetric stretch and $C_\alpha - C'_\alpha$ stretch, respectively [35]. The peak at 1095 cm⁻¹ corresponds to the C–O–C deformation, while the peaks between 440 cm⁻¹ and 991 cm⁻¹ are related to the oxyethylene ring deformation of PEDOT [35–37]. The decrease in the overall band intensity of the sample polarized at –1.0 V shows the loss of SDS due to the dedoping process.

In Figure 11, the peaks observed at 1417 cm⁻¹ and 1432 cm⁻¹ correspond to the neutral (quinoid) $C_\alpha = C_\beta$ and oxidized (benzoid) $C_\alpha - C_\beta$ symmetric stretching modes, respectively. The undoped PEDOT film (0 V) is highly oxidized as it contains the highest amount of the benzoid (oxidized) structures (Figure 11a). The oxidation level decreases as the polymer film is polarized from –0.2 to –0.6 V, demonstrating the dedoping process of the CP on the application of cathodic potentials (Figure 11c,d). The band shift from 1424 to 1417 cm⁻¹ (redshift) in the spectra of polarized samples on stepping to higher cathodic potentials indicates the change in the doping level, as the benzoid-predominated PEDOT (at 0 V) transforms to a quinoid-structure-dominant polymer film at –0.6 V and –1.0 V [35–37]. The percentage of the benzoid structure in the polymer increases to approximately 40% (Table 4) after the sample was polarized at –1.0 V (in agreement with the EIS data) to accelerate nucleophilic attacks (by $O_2$ and OH⁻ ions) on the PEDOT chains due to excessive oxygen reduction at higher cathodic potentials.

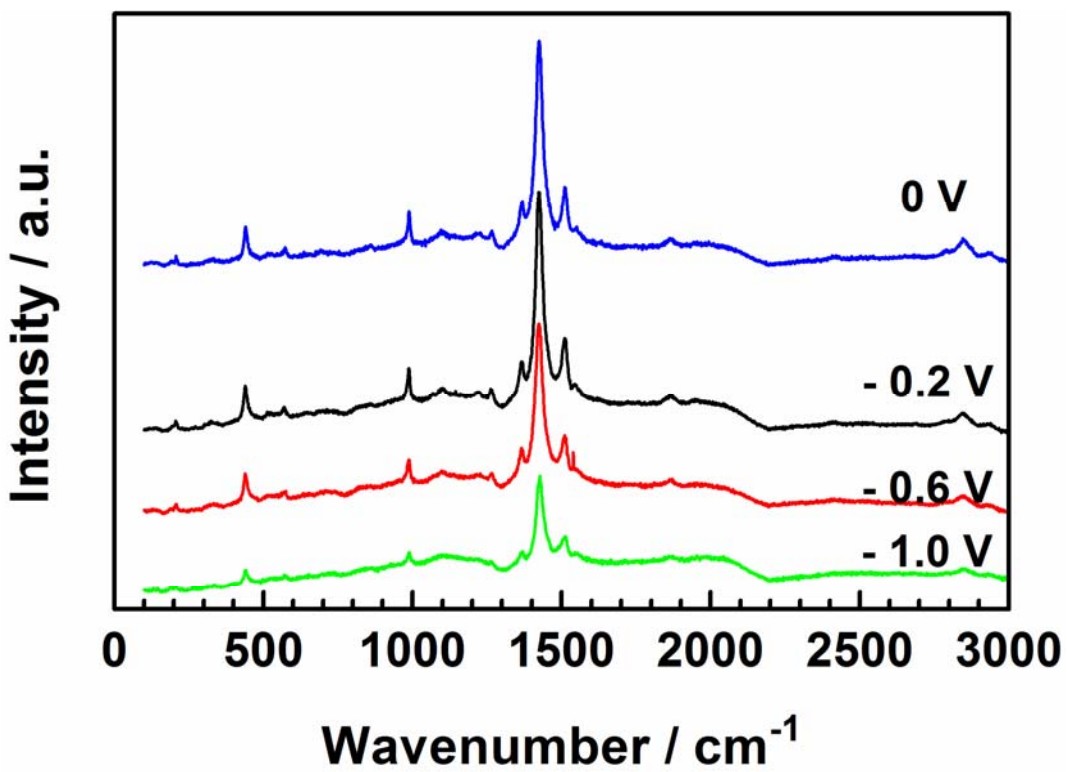

**Figure 10.** Raman spectra of PEDOT polarized at each potential for 300 s in NaCl solution.

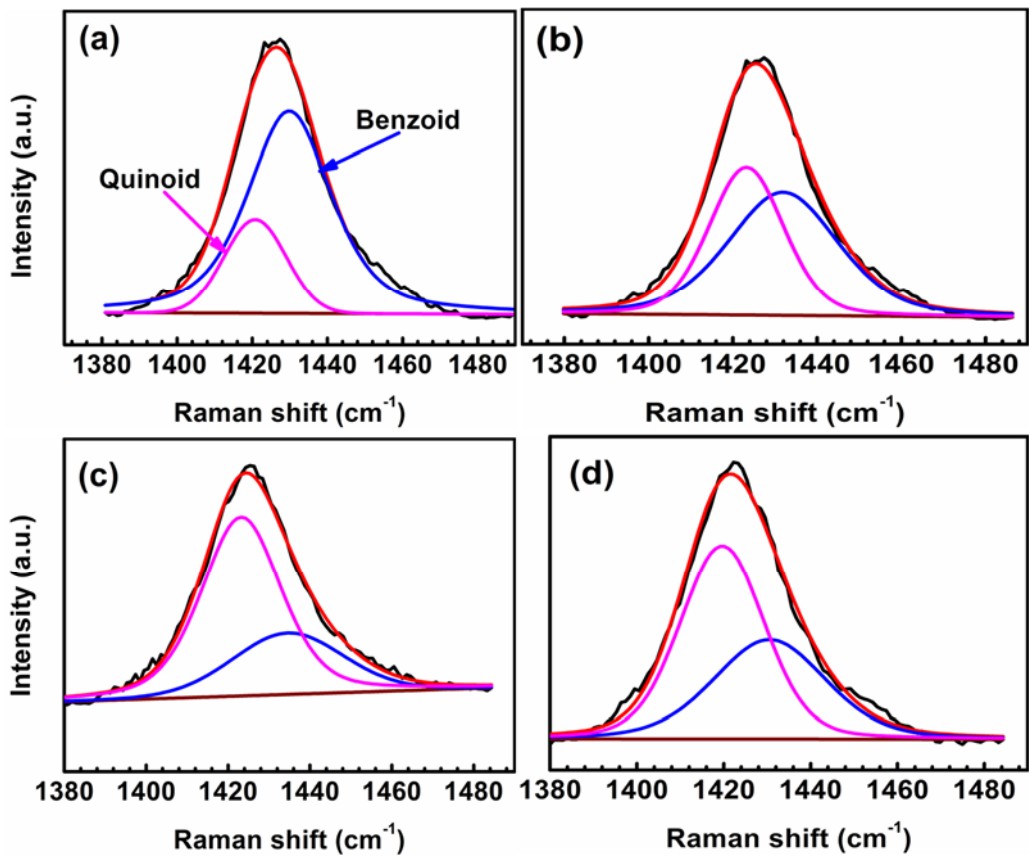

**Figure 11.** Micro-Raman spectra showing the neutral (quinoid) and oxidized (benzoid) regions of the PEDOT film polarized at 0 V (**a**), −0.2 V (**b**), −0.6 V (**c**), and −1.0 V (**d**) in NaCl solution. The black and red lines represent the raw data and fitted lines, respectively.

**Table 4.** Percentage composition of the oxidized (benzoid) structures in the polarized PEDOT film.

| Polarization Potential | 0 V | −0.2 V | −0.6 V | −1.0 V |
|:---:|:---:|:---:|:---:|:---:|
| Percentage | 77% | 46% | 34% | 40% |

*3.7. Surface Characterization*

Figure 12 shows the surface of the undoped PEDOT film examined by SEM after 60 h of corrosion tests in NaCl solution. The PEDOT film exhibits non-uniform porosity with swellings and micro-cracks (Figure 12c,d), characteristic of PEDOT film in an over-oxidized state. Corrosion products are observed on the PEDOT film, which evolves from the substrate via the coating pores (Figure 12c). EDS analysis of the spot marked by a red rectangle on the PEDOT film (Figure 12d) shows that the elemental composition in weight % includes 67.3 C, 22.4 O, 7.20 S, 1.11 Na, 0.91 Cl, 0.48 Fe, 0.33 Cr, and 0.27 Mo. The high S-content is attributed to SDS dopant anions in the polymer matrix, which are released during the reduction of PEDOT by galvanic coupling with the steel substrate (Figures 5 and 6).

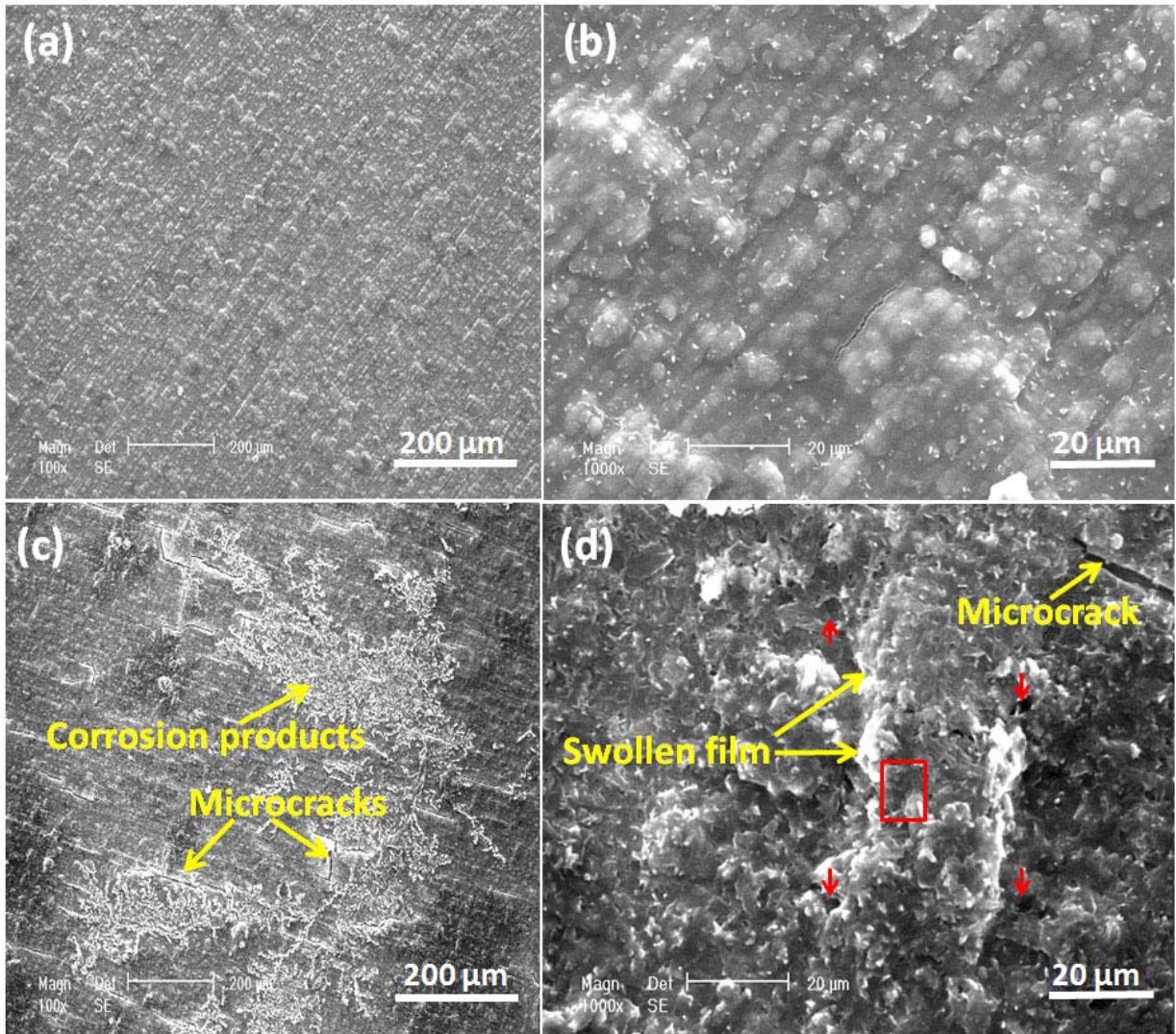

**Figure 12.** SEM images of freshly prepared PEDOT coating (**a,b**) and after corrosion immersion tests in 3.5% NaCl solution (**c,d**). Red arrows mark the pores on the PEDOT film, while the red box represents the area of the surface analyzed by EDS.

*3.8. XPS*

The chemical states of tested PEDOT films were further examined by XPS analysis. Both thiophenes and sulfonates are detected in the S 2p spectra of PEDOT (Figure 13a). The sub-peaks located at 168.5 and 169.6 correspond to the sulfonate components of the SDS dopant, while the sub-peaks located at 163.7 and 164.6 are attributed to the S of the thiophene rings (C–S–C) [14]. The peak at 165.4 eV is assigned to the oxidized thiophene sulfur (with a positive partial charge, $S^{\delta+}$) due to polarons and bipolarons [30,31]. The peak at 169.0 eV indicates sulfur oxidation in the PEDOT thiophene ring [30]. S=O groups can form after prolonged exposure to dissolved oxygen due to oxidation of the thiophene sulfur [29–33].

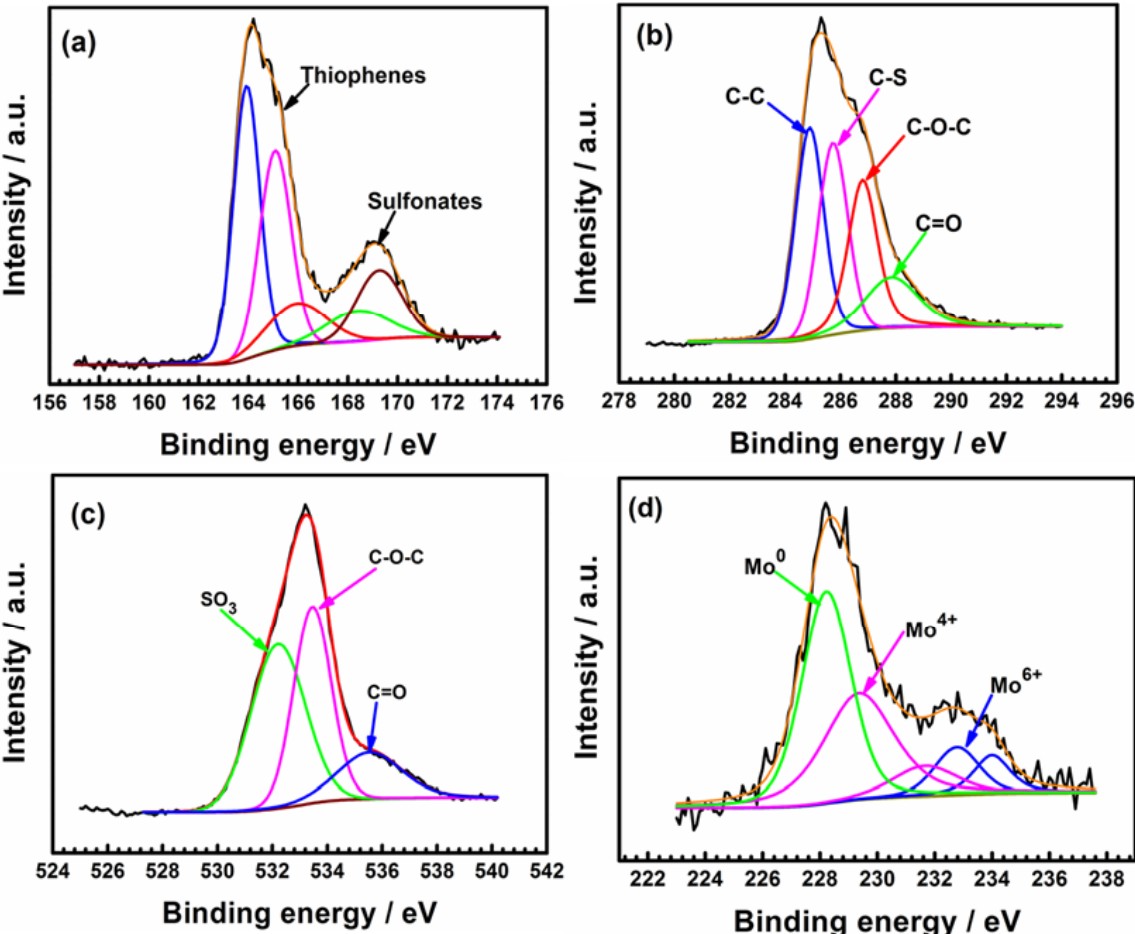

**Figure 13.** XPS spectra of C 1s (**a**), S 2p (**b**), O 1s (**c**), and Mo 3d (**d**) of PEDOT coating after corrosion tests in 3.5% NaCl solution at open circuit potential. The black and red lines represent the raw data and fitted lines, respectively.

In the C 1s spectrum (Figure 13b), the intense peak at 285.6 eV corresponds to carbon atoms belonging to C–C and C–H groups at α and β positions of the thiophene rings [28]. The peak at 285.6 eV corresponds to the C–S bonds of thiophene and the C–C bonds of the SDS anions [14,38]. A fourth peak in the C 1s spectrum at 287.9 eV and the O 1S signal at 535.7 eV (Figure 13b,c) indicate the formation of C=O groups associated with the degradation of the polymer film [33]. The formation of the C=O bond implies the breaking of the π-conjugation of the thiophene ring [29,32,33]. DO in the test media can attack the PEDOT chain during the prolonged immersion test period to promote their degradation [34–37]. Similar degradation mechanisms have been reported in the references [34–37].

Figure 13d shows that Mo enriched the oxide layer of the steel substrate. The spectrum of Mo 3d (Figure 13d) indicates the presence of $Mo^0$, $Mo^{4+}$, and $Mo^{6+}$. The two peaks at 232.87 eV and 235.99 eV are attributed to $MoO_4^{2-}$ and $MoO_3$, respectively [39]. It is well known that $Mo^{6+}$ promotes the formation of $MoO_3$ and $MoO_4^{2-}$ in the passive film of 316L SS. The Mo-enriched oxide layer hinders ion diffusion to the steel substrate, resulting in the increased resistance of coated electrode after 60 h of testing (Table 2).

## 4. Discussion

The effect of the redox switch and coupling on the anti-corrosion properties of PEDOT film in NaCl solution is discussed under the following subheadings: corrosion protection mechanism and polarization behavior of the PEDOT film, as depicted in Figure 14.

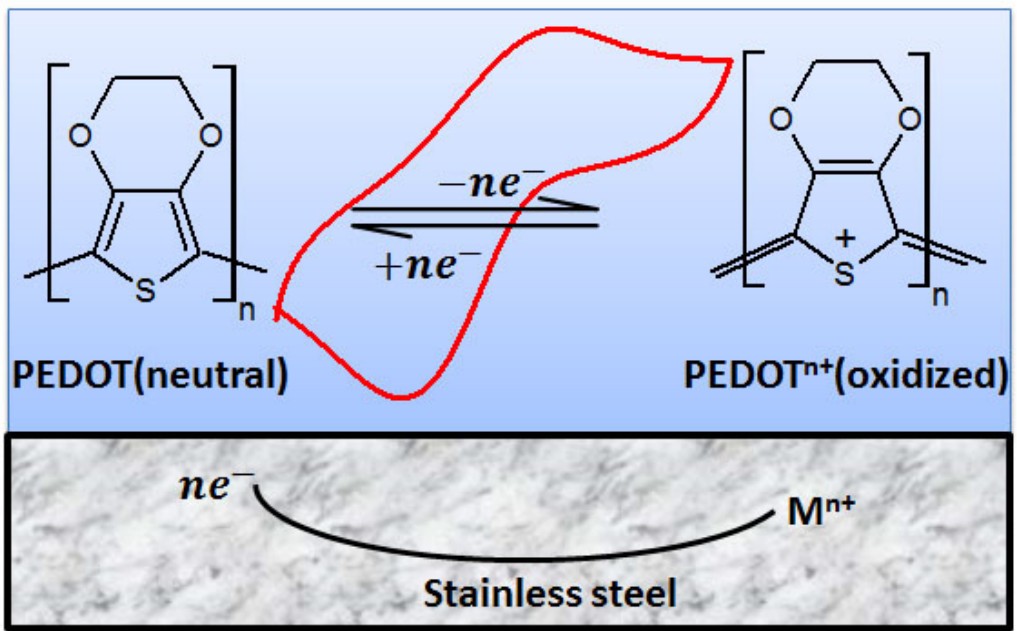

**Figure 14.** Proposed mechanism for the redox behavior of PEDOT film immersed in 3.5% NaCl solution.

### 4.1. Corrosion Protection Mechanism

The electrochemical polymerization of EDOT on SS generates positive charge carriers, which means the freshly prepared PEDOT film is in the oxidized state ($PEDOT^{n+}$). When the electrolyte reaches the steel surface, anodic substrate oxidation occurs, which yields electrons that partially reduce/discharge PEDOT at the cathode (Figures 4 and 5). At the same time, cations ($M^{n+}$) evolve to react with the corrosive medium to form an oxide layer [27,38–40]:

Anodic reactions:

$$M \longrightarrow M^{n+} + ne^- \tag{2}$$

Cathodic reactions:

$$PEDOT^{n+} + ne^- \longrightarrow PEDOT \tag{3}$$

The discharged PEDOT in reaction (3) is later recharged/reoxidized by mediating ORR:

$$\frac{n}{4}O_2 + \frac{n}{2}H_2O + ne^- \longrightarrow nOH^- \tag{4}$$

$$PEDOT + \frac{n}{4}O_2 + \frac{n}{2}H_2O \longrightarrow PEDOT^{n+} + nOH^- \tag{5}$$

$$M^{n+} \frac{n}{4}O_2 + \frac{n}{2}H_2O \longrightarrow M(OH)_n \tag{6}$$

Reactions (1) and (4) will couple to produce $M(OH)_n$, which represent the unstable hydroxides of Fe or Mo that are converted to more stable $Fe_3O_4/Fe_2O_3$ and $MoO_3/MoO_4^{2-}$ in the oxide layer, which hinders the diffusion of corrosive anions to the steel substrate at the coating pores and defects (Figures 12 and 13).

The reduction/dedoping of PEDOT by galvanic coupling with the steel substrate could initiate a self-healing process in the scratched area of the coating by releasing SDS anions, which cannot diffuse out of the polymer matrix, due to its large size (Figure 11). The dopant anions may react with metal ions to form a "complex" (reaction (7)):

$$\{PEDOT^{n+}/SDS^-\} + e^- + M^+ \rightleftharpoons \{PEDOT/SDS^-/M^+\} \tag{7}$$

The on-demand release of inhibitors is exploited in smart coatings to heal defective areas on coatings. Documented reports show that an aqueous SDS solution inhibits steel corrosion in acidic and alkaline solutions [41,42]. Dealloyed cations can also chelate with the CP via the thiophene sulfur to form $PEDOT/SDS/M^+$ "complexes" [38].

Furthermore, Figure 2 shows that PEDOT exhibits excellent stability after 20 charging–discharging cycles, which is attained mainly by the charge compensation phenomenon. Counterions, $X^-$ (mainly $OH^-$ and $Cl^-$), are incorporated into the polymer film from the electrolyte upon reoxidation to form the $PEDOT^+/SDS^-/M^+/X^-$ "complex" and maintain charge neutrality [42]:

$$\{PEDOT/SDS^-/M^+\} + nX^- \rightleftharpoons \{PEDOT^{n+}/SDS^-/M^+/X^-\} \tag{8}$$

The XPS and EDS results (Figure 13) demonstrate that reactions (7) and/or (8) are favored during the test.

By mediating ORR, the PEDOT film limits the amount of oxygen available at the surface of SS (reaction (8)). However, the fluctuations in the potential of the coated electrode (Figures 4 and 5) show that reactions (2)–(6) influenced the stability and performance of the PEDOT film. The Nernst equation expresses the $E_{OCP}$ of CPs [43]:

$$E_{OCP} = E^0 + \frac{RT}{F}\ln\frac{[PEDOT^{n+}]}{[PEDOT]} + \Delta\varphi_D \tag{9}$$

where $E^0$ is the standard potential of the redox system; $[PEDOT^{n+}]$ and $[PEDOT]$ are the concentrations of the oxidized (benzoid) and neutral (quinoid) forms of PEDOT, respectively. The standard gas constant is denoted as $R$, T is the temperature, $F$ is the Faraday constant, and $\Delta\varphi_D$ is the Donnan potential at the polymer/solution interface.

The charging–discharging processes of the PEDOT film affect its potential by changing the ratio of $[PEDOT^{n+}]/[PEDOT]$. As the potential stabilizes, the rates of charging and discharging on the polymer film become equal [38,42]:

$$K_{OX}[O_2]^{n/4}[PEDOT] = K_{red}[PEDOT^{n+}][OH^-]^n \tag{10}$$

where $K_{ox}$ and $K_{red}$ are rate constants of the oxidation and reduction reactions, respectively. $[O_2]$ and $[OH^-]$ are the concentrations of $O_2$ and $OH^-$, respectively:

$$\frac{[PEDOT^{n+}]}{[PEDOT]} = \frac{K_{ox}[O_2]^{n/4}}{K_{red}[OH^-]^n} \tag{11}$$

From relationship (11), it can be deduced that $[O_2]$ and $[OH^-]$ are important factors that influence the stability and conductivity evidenced by the $E_{OCP}$ of PEDOT. Prolonged exposure of the PEDOT film to $O_2$ and $OH^-$ in NaCl solution results in the formation of C=O groups associated with the over-oxidation (degradation) of the polymer film (Figure 4). The non-uniform porosity of the over-oxidized PEDOT film could also create gradients of molecules (e.g., $O_2$, $H_2O$) and ions ($Cl^-$, $OH^-$), resulting in the formation of micro-galvanic cells, which could enhance the corrosion process of the steel substrate (Figure 12).

*4.2. Polarization Behavior*

PEDOT exhibits a p-type semi-conductor (i.e., holes extracting) characteristic [9,11], which implies that the polymer film could hinder electron transport [27]. Interactions between the polymer unit segment and the neighboring oxide layer on SS lead to the formation of electronic bands, which are separated by an energy gap (Figure 6). Charge transportation through the oxide layer to the PEDOT film and vice versa may be difficult owing to the difference in the position of their valence bands and the energy gaps (Figure 8). During electrochemical doping, ions are incorporated into or extracted from the PEDOT film (reactions (7) and (8)). The band gap energy is effectively minimized due to electrons transferred from redox reactions on the polymer, which manifests as improved electrical conductivity (Figures 7, 8, 10 and 11). The conductivity increase in these materials arises from the introduction of polaronic (cation) and/or bipolaronic (dication) species into the polymer backbone, which are charge-compensated by counterions [43].

When the PEDOT film is polarized at sufficiently high cathodic potentials (from –0.4 V to $-0.7$ V), the polarization of the polymer in this potential regime causes several changes in the polymer. PEDOT is dedoped, with the release of dopant (SDS) anions, while cations in the test solution (e.g., $Na^+$) are inserted for charge compensation. In addition, the PEDOT volume will shrink to close ion channels [28,29], thus making the invasion of corrosive anions difficult (Figures 7 and 9). The percentage of benzoid structures in the PEDOT film decreases to mitigate attacks by nucleophiles (mainly $OH^-$ ions and intermediate products are formed from ORR), which can cause degradation of the PEDOT film by breaking the $\pi$-conjugation of the thiophene ring. Alternatively, ORR may directly consume the electrons on the PEDOT film, a pathway that does not involve dedoping the polymer [27]. However, when PEDOT is polarized at cathodic potentials >–0.7 V, the polymer is deeply reduced/dedoped, and more $OH^-$ ions are generated from the excessive oxygen reduction in this potential range. Fast ejection of the abundant $OH^-$ ions generated by the high applied force causes a loss of electroactivity and possibly deformation of the PEDOT chain (Figures 7 and 9). In addition, the increased percentage of the oxidized quinoid structures on the PEDOT chains (Table 4) could promote nucleophile attack on the polymer.

In summary, the PEDOT film is reduced by coupling with redox reactions on the SS substrate, while evolved cations react with $OH^-$ to form an insoluble oxide layer enriched by Mo species. Whereas the PEDOT film becomes cathodic, the steel region under the rust layer is anodic. PEDOT is reoxidized by mediating ORR, which limits the amount of oxygen available at the substrate while the porosity of the coating increases, thus opening up ion channels in the coating to attacks by corrosive anions. Attacks on the polymer chain by $OH-$ ions generated from ORR cause degradation of the polymer film after prolonged exposure to dissolved oxygen. When PEDOT is polarized at sufficiently high cathodic potentials (ranging from –0.4 V to –0.7 V), the polymer is dedoped, and the volume shrinks to prevent nucleophile attacks. There is a need for an additional reservoir of negative charges to be integrated into the coating to supply electrons for reduction and sustain the CP in the reduced state, which offers better protection of the steel substrate.

## 5. Conclusions

Strongly adherent PEDOT film was electrodeposited on stainless steel (SS), and redox interactions of the CP-coated electrode were examined in NaCl aqueous solution at various polarization potentials. The following conclusions can be inferred:
(1) The PEDOT film exhibited barrier protection and mediated the oxygen reduction reaction on SS.
(2) The PEDOT film was initially reduced by coupling with the SS substrate and then re-oxidized by dissolved $O_2$.
(3) The scratched PEDOT film prevented surface charge localization, resulting in the electrochemical protection of SS.

(4) With polarization at cathodic potentials from –0.4 V to –0.7 V, PEDOT was dedoped, which caused the polymer structure to shrink, thereby preventing nucleophile attacks.

(5) The electroactivity and conductivity of the polymer film declined when PEDOT was polarized at potentials >–0.7 V.

(6) $OH^-$ ions generated from the oxygen reduction reaction attacked the polymer chain, causing the formation of carbonyl and sulfone groups, evidenced by XPS results of the degraded film. Degradation (over-oxidation) of the PEDOT coating due to attacks by $OH^-$ ions was favored by prolonged exposure to oxygen.

**Author Contributions:** Conceptualization, M.Y.; methodology, M.Y.; validation, M.Y. and F.L.; formal analysis, V.M.U.; investigation, V.M.U.; resources, M.Y.; data curation, V.M.U.; writing—original draft, V.M.U.; writing—review and editing, V.M.U., M.Y. and F.L.; visualization, V.M.U.; supervision, F.L. and E.-H.H.; project administration, M.Y.; funding acquisition, M.Y. All authors have read and agreed to the published version of the manuscript.

**Funding:** This work was supported by the "National Natural Science Foundation of China" (No. 52071320).

**Institutional Review Board Statement:** Not applicable.

**Informed Consent Statement:** Not applicable.

**Data Availability Statement:** Not applicable.

**Acknowledgments:** The authors are grateful to Bright Okonkwo for improving the grammar in the manuscript and Gao Bowen for technical support during the laboratory experiments.

**Conflicts of Interest:** The authors declare no conflict of interest.

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
