# Peer review of "Effect of Redox Switch, Coupling, and Continuous Polarization on the Anti-Corrosion Properties of PEDOT Film in NaCl Solution"

_coatings, doi:10.3390/coatings13050944_

Round 1
Reviewer 1 Report
Reviewer Recommendation and Comments for manuscript coatings-2287726 with the title: “Redox and Corrosion Behavior of Poly(3,4-Ethylenedioxythiophene) Film on Stainless Steel in NaCl Solution”, authors: V.M. Udowo, M. Yan, F. Liu, E. Han.
The authors report the electrochemical synthesis of poly(3,4-ethylenedioxythiophene = PEDOT) film on 316L stainless steel by galvanostatic electrodeposition. The authors study the redox activities of the PEDOT film and the interactions between the film and the SS substrate by electrochemical and spectroscopic analyzes in 3.5% NaCl solution.
The main comments that I find useful for improving the quality of the article are presented below:
The manuscript must be proofread by a native English speaker.
*│The journal template must be used.
*line 39│”Yan et al. [6–7]”. Yan et al. correspond to [8] and [9].
*│All references must be checked and corrected.
*line 41│”Torresi et al. [5] found that”. Reference [5] do not correspond to Torresi.
*line 47│”In fact, Oxygen has a”. The typo must be corrected.
*line 56│”Zhang et al. [12] and Gao et al. [13] reported the”. Zhang and Gao do not correspond to [12] and [13].
*line 59│”Zhu et al. [15] demonstrated”. Zhu do not correspond to [15].
*line 90│”potential scan rate of 0.10 mV/S.”. The potential scan rate of 0.1 mV/s is too low for cyclic voltammetry. (s should be used instead of S). It must be explained why such a low rate was used.
*line 117│”The details of the XPS analytical process can be found in the reference [18].”. Reference [18]: Wu, T. Xu, J. Sun, C. Yan, M. Yu, C. Ke, W. Microbiological corrosion of pipeline steel under yield stress in soil environment, Corros. Sci. 2014, 88, 291 305. ? Which reference should be studied?
*line 120 to 124│Why are different sizes used for text?
*line 120│”mechanism of growth”. Figure 1b cannot indicate the growth mechanism of the electrodeposited film. The nucleation mechanism (2D/3D - instantaneous/progressive is not indicated by figure 1b).
*line 126│”(Figure 1b)”?
*line 129│”(Figure 1b)”?
*line 130│”We assume that the electropolymerization of the EDOT monomer is a bi-electronic process”. What is the scientific basis for this assumption? According to Figure 1b, the current can be attributed to both electropolymerization and other processes. Considering that the electrode potential varies between 1.2 and 1.7 V, in this potential range other species can participate in electrode processes. Under these conditions, figure 1b, for example, must also contain the chronopotentiogram corresponding to the supporting electrolyte (same electrolyte solution but without monomer). Why is the process bi-electronic? The electropolymerization process can start at 1.2 V, and at potentials higher than 1.6 V, the polymer passes into the oxidized state. In the considered range, two processes take place: electropolymerization and electrooxidation.
*line 131│”with 100% current efficiency.”? In this case, the consideration can only be purely theoretical.
*line 140│”The reduction peak appeared at –0.2 V, while the oxidation peak appeared at –0.6 V (Figure 2).” Isn't it the other way around? (the oxidation peak at -0.2 V and the reduction peak at -0.6 V)? To eliminate any confusion, the authors should specify how the potential is generated.
*line 142│”The peak separation shows that the redox reactions on the polymer are reversible and have fast kinetics”. It is observed that the current density corresponding to the peak at -0.6 V decreases as the number of cycles increases. A first conclusion that the peak is at most quasi-reversible.
*3.2. Open circuit potential │Does the open circuit potential correspond to the two electrodes in the galvanic couple or not? It is not clear whether the two electrodes are connected or not. Is the galvanic coupling described in section 3.3?
*line 197│”lowering impedance the impedance” The typo must be corrected.
*lines 225-228│different font?
*line 242│”(Figure 10).”?
*line 248│”backbone is dominated by the oxidized structure at higher cathodic potentials”. The authors state that as cathodic potentials increase, the polymer chain is more oxidized? Is the electrode polarized anodic or cathodic??
*3.5. Surface Characterization │How do you explain the very high sulfur content knowing that 316L stainless steel contains up to 0.03% S? How do you explain the lack of nickel knowing that 316L stainless steel contains up to 18% Ni?
*line 291│”The corrosion mechanism of corrosion of PEDOT”. The manuscript must be proofread by a native English speaker.
*line 303│The stoichiometry of reaction 5 must be corrected.
*line 337│”(Figure 12).”?
*[PEDOTn+]│
*│How is "n+" determined? The polymer is not structurally characterized. The structure is not known. What is the molecular mass? What is the degree of polymerization? Are there side chains? The polymer is presented in an idealistic manner. It is more accurate to present the results as the effect of a polymer film rather than as a polymer that cannot be characterized. What is the structure of the polymer if XPS analysis indicates the presence of C=O bonds, in NaCl solution, at OCP? This bond, C=O, is not in Figure 1a.
*│It could be a confusion between " Acknowledgments " and "Funding". Must be checked.
*The Coatings journal require a specific format of references, authors must pay more attention in their writing. No reference is written according to the format required by the journal.
*There are some grammar and typing mistakes.
*The authors must revise the entire manuscript.
Author Response
We express gratitude for the many instructive comments on our Manuscript ID coatings-2287726. The comments are precious for us to improve this manuscript for publication. Revisions made to the manuscript in response to all the comments are listed in a separate file titled “Response to Reviewers' comments”. Major changes to the manuscript are marked in blue in the revised manuscript, and the page numbers cited below refer to the revised manuscript. Since revisions are also made to some figures, the revised figures have also been resubmitted.
We hope the responses adequately addressed all the suggestions and questions posed. Thank you again for your assistance in preparing our manuscript for publication.
Response to Reviewer(s)' Comments:
Reviewer: 1
1. The manuscript must be proofread by a native English speaker. The journal template must be used.
We have revised the whole manuscript concerning this recommendation.
2. Line 39│”Yan et al. [6–7]”. Yan et al. correspond to [8] and [9].
Corrected.
- All references must be checked and corrected.
Corrected.
- Line 41│”Torresi et al. [5] found that”. Reference [5] do not correspond to Torresi.
Corrected.
- Line 47│”In fact, Oxygen has a”. The typo must be corrected.
Corrected.
- Line 56│”Zhang et al. [12] and Gao et al. [13] reported the”. Zhang and Gao do not correspond to [12] and [13].
Corrected.
- Line 59│”Zhu et al. [15] demonstrated”. Zhu do not correspond to [15].
Corrected.
- Line 90│”potential scan rate of 0.10 mV/S.”. The potential scan rate of 0.1 mV/s is too low for cyclic voltammetry. (s should be used instead of S). It must be explained why such a low rate was used.
The unit of scan rate is “10 mV/s or 0.10 V/s” not “0.10 mV/s”. The explanation for the slow scan rate has been added to the manuscript:
The slow potential scan rate was chosen to allow sufficient time for the products of the reduction reaction in the forward scan to participate in the backward reaction and generate clear anodic and cathodic reversible peaks.
- line 117│”The details of the XPS analytical process can be found in the reference [18].”. Reference [18]: Wu, T. Xu, J. Sun, C. Yan, M. Yu, C. Ke, W. Microbiological corrosion of pipeline steel under yield stress in soil environment, Corros. Sci. 2014, 88, 291 305. ? Which reference should be studied?
Reference [18]: Wu, T. Xu, J. Sun, C. Yan, M. Yu, C. Ke, W. Microbiological corrosion of pipeline steel under yield stress in soil environment, Corros. Sci. 2014, 88, 291 305.
- Line 120 to 124│Why are different sizes used for text?
Corrected.
- Line 120│”mechanism of growth”. Figure 1b cannot indicate the growth mechanism of the electrodeposited film. The nucleation mechanism (2D/3D - instantaneous/progressive is not indicated by figure 1b).
This sentence has been corrected in section 3.1, paragraph 1:
The potential-time plot reveals the nucleation of process of PEDOT film on SS. Figure 2a presents two distinct regions. The electropolymerization process starts around 1.2 V, and at potentials higher than 1.6 V, the polymer passes into the oxidized state (Figure 2a).
- Line 126│”(Figure 1b)”?
Corrected.
- Line 129│”(Figure 1b)”?
Corrected.
- line 130│”We assume that the electropolymerization of the EDOT monomer is a bi-electronic process”. What is the scientific basis for this assumption? According to Figure 1b, the current can be attributed to both electropolymerization and other processes. Considering that the electrode potential varies between 1.2 and 1.7 V, in this potential range other species can participate in electrode processes. Under these conditions, figure 1b, for example, must also contain the chronopotentiogram corresponding to the supporting electrolyte (same electrolyte solution but without monomer). Why is the process bi-electronic? The electropolymerization process can start at 1.2 V, and at potentials higher than 1.6 V, the polymer passes into the oxidized state. In the considered range, two processes take place: electropolymerization and electrooxidation. Line 131│”with 100% current efficiency.”? In this case, the consideration can only be purely theoretical.
Calculation of the polymer film thickness has been removed from the paper since the information contributed little to the overall discussion in the manuscript.
- line 140│”The reduction peak appeared at –0.2 V, while the oxidation peak appeared at –0.6 V (Figure 2).” Isn't it the other way around? (the oxidation peak at -0.2 V and the reduction peak at -0.6 V)? To eliminate any confusion, the authors should specify how the potential is generated.
This sentence has been corrected in section 3.2, paragraph 1, which now reads:
By visual inspection of the CV curve, the oxidation peak is observed in the potential range from –0.3 V to –0.1 V, while the reduction peak appeared between –0.5 V and –0.7 V.
- Line 142│”The peak separation shows that the redox reactions on the polymer are reversible and have fast kinetics”. It is observed that the current density corresponding to the peak at -0.6 V decreases as the number of cycles increases. A first conclusion that the peak is at most quasi-reversible.
The following sentence in Line 142 of section 3.2:
The peak separation and the decrease in the current density of the reduction peak (around −0.6 V) as the number of cycles increases indicates that the redox process on the polymer is quasi-reversible
- 3.2. Open circuit potential │Does the open circuit potential correspond to the two electrodes in the galvanic couple or not? It is not clear whether the two electrodes are connected or not. Is the galvanic coupling described in section 3.3?
The open circuit potential corresponds to the two electrodes electrically coupled. However, the differences are explained in section 3.4, Paragraph 3. Section 3.4, Paragraph 1 describes the galvanic coupling results:
The PEDOT–coated electrode (WE) and the bare SS (CE) were electrically coupled in NaCl solution via a potentiostat in ZRA mode, resulting in a steady coupling current of about 0.012 μA (Figure 5). The mixed potential (Emix) of PEDOT decreased from −0.060 to −0.168 V in 6 hours (Figure 5), demonstrating the reduction of the polymer to its semi-conducting form. The extent of reduction is determined by the potential drift of the bare SS exposed to corrosion reactions in NaCl solution. After 6 hours, an inflection is observed, and then the Emix ennobles significantly (Figure 5), suggesting reoxidation of the CP by coupling with oxygen reduction reaction on the polymer film. Fluctuations in the mixed potential of PEDOT during coupling (Figure 5–6) suggest ionic exchanges occurring between the CP film and the electrolyte.
- Line 197│”lowering impedance the impedance” The typo must be corrected.
Corrected.
- Lines 225-228│different font?
Corrected.
- Line 242│”(Figure 10).”?
Corrected.
- Line 248│”backbone is dominated by the oxidized structure at higher cathodic potentials”. The authors state that as cathodic potentials increase, the polymer chain is more oxidized? Is the electrode polarized anodic or cathodic??
The sentence has been corrected in section 3.5, Paragraph 1:
The decrease in the overall band intensity of the sample polarized at –1.0 V shows the loss of SDS due to the dedoping process.
- 3.5. Surface Characterization │How do you explain the very high sulfur content knowing that 316L stainless steel contains up to 0.03% S? How do you explain the lack of nickel knowing that 316L stainless steel contains up to 18% Ni?
The high sulfur content of the polymer film is due to the release of dodecyl sulfonate ions when the polymer is dedoped by electrical stimulation or galvanic coupling with the steel substrate during the test in the NaCl solution. Leaching of elements during surface treatment and cation expulsion during redox processes on the polymer film may be the reason for the non-detection of Ni by EDS.
- Line 291│”The corrosion mechanism of corrosion of PEDOT”. The manuscript must be proofread by a native English speaker.
We have corrected the sentence, and a native English speaker revised the whole manuscript.
- Line 303│The stoichiometry of reaction 5 must be corrected.
The stoichiometry of reactions 5-6 is corrected:
O2 + H2O + ne− nOH− (5)
PEDOT + O2 + H2O PEDOTn+ + nOH− (6)
- Line 337│”(Figure 12).”?
Corrected.
- [PEDOTn+]│How is "n+" determined? The polymer is not structurally characterized. The structure is not known. What is the molecular mass? What is the degree of polymerization? Are there side chains? The polymer is presented in an idealistic manner. It is more accurate to present the results as the effect of a polymer film rather than as a polymer that cannot be characterized. What is the structure of the polymer if XPS analysis indicates the presence of C=O bonds, in NaCl solution, at OCP? This bond, C=O, is not in Figure 1a.
The structure of the polymer is given in Fig. 1. PEDOTn+ represents the oxidized form of the polymer. n+ is not a fixed value because the oxidized form of the polymer may exist as bipolaron (dication) and polaron (cation). The C=O group detected in the XPS result is traced scissoring of the polymer chain by hydroxide ion attack, which is also known as degradation (over-oxidation). Similar degradation mechanism of the PEDOT chain has been reported in references 9, 29,35-37.
28.│It could be a confusion between " Acknowledgments " and "Funding". Must be checked.
Corrected.
- The Coatings journal require a specific format of references, authors must pay more attention in their writing. No reference is written according to the format required by the journal.
The references have been formatted according to the style required by the journal.
- There are some grammar and typing mistakes. The authors must revise the entire manuscript.
We have revised the whole manuscript with regards to this recommendation.

Reviewer 2 Report
The manuscript presents an interesting study about the corrosion behaviour of PEDOT film deposited on the surface of stainless steel. However, the paper needs major revisions before it is processed further, some comments follow:
Abstract
The abstract must be improved. Please highlight the novelty and importance of this study. Also, add some quantitative results.
Materials and methods
Add a table with the chemical composition of the used steel.
Results
The figures must be added in the manuscript text body, exactly where are discussed. Also, the tables.
Conclusions
Add quantitative discussion, suggestions and limitations.
Author Response
Response to Reviewer 2 Comments
- The abstract must be improved. Please highlight the novelty and importance of this study. Also, add some quantitative results.
We have revised the abstract:
Conjugated poly(3,4-ethylenedioxythiophene) (PEDOT) film was electrochemically synthesized on stainless steel (SS). Redox interactions between the PEDOT film and the SS substrate were examined in 3.5 wt. % NaCl aqueous solution with the aid of electrochemical and spectroscopic analyses. The results show that the PEDOT film exhibited a barrier effect and mediated oxygen reduction reaction, thus hindering ion diffusion to the steel substrate. Localized electrochemical impedance spectroscopy (LEIS) of the scratched area on the polymer film shows that PEDOT healed the defect by coupling with redox reactions on the steel surface to prevent charge localization and concentration. The electroactivity of the polymer film declined when PEDOT was polarized at potentials > –0.7 V. Prolonged exposure of the PEDOT film to dissolved oxygen in NaCl solution resulted in the polymer's over–oxidation (degradation), evidenced by the formation of a carbonyl group in the spectroscopic result. The degradation of PEDOT was attributed to chain scissoring due to hydroxide ion attacks on the polymer chain.
- Materials and methods: Add a table with the chemical composition of the used steel.
Table 1, containing the chemical composition of the used steel, is included in the manuscript.
- Results: The figures must be added in the manuscript text body, exactly where are discussed. Also, the tables.
Corrected.
- Conclusions: Add quantitative discussion, suggestions and limitations
The conclusion has been revised with regard to this recommendation.
Reviewer 3 Report
The title of the manuscript is very poor and doesn’t reflect the content of the work.
The manuscript text is very badly formatted and absolutely don’t meet the requirements for submitted manuscripts.
In many places, there is no connection between the text and drawings, citations and text, figures and figure captions (e.g., fig. 1).
There are chemical mistakes and wrong equations (e.g., Fe2O3 is known not to be protective species).
The novelty and relevance of the work are poor formulated. The purpose of the investigation is also very vague (lines 62-67). What is the difference between already published and presented data?
The data obtained are discussed very superficially. It concerns, in particular, the mechanism of anticorrosive action of oligomer films. It is absolutely unclear what happens to the coating under electrochemical conditions. Fig. 13 is also meaningless and primitive.
Author Response
Response to Reviewer 3 Comments
- The title of the manuscript is very poor and doesn’t reflect the content of the work.
The title of the manuscript has been revised to reflect the content of the work:
Effect of redox switch, coupling and continuous polarization on the anti-corrosion properties of PEDOT film in NaCl solution
- The manuscript text is very badly formatted and absolutely don’t meet the requirements for submitted manuscripts.
We have revised the whole manuscript concerning this recommendation.
- In many places, there is no connection between the text and drawings, citations and text, figures and figure captions (e.g., fig. 1).
The text has been corrected to match with the right citations, figures and drawings.
- There are chemical mistakes and wrong equations (e.g., Fe2O3 is known not to be protective species).
This statement is corrected in the last paragraph of sections 3.7 and 4.2:
The oxide layer of 316L stainless steel is enriched by molybdenum oxide species, which boosts its protectiveness.
- The novelty and relevance of the work are poor formulated. The purpose of the investigation is also very vague (lines 62-67). What is the difference between already published and presented data?
Paragraph 4 of section 1 has been revised to address this concern.
- The data obtained are discussed very superficially. It concerns, in particular, the mechanism of anticorrosive action of oligomer films. It is absolutely unclear what happens to the coating under electrochemical conditions. Fig. 13 is also meaningless and primitive.
Sections 4.1 and 4.2 of the discussion have been revised to outline what happens to the coating under electrochemical conditions.
Round 2
Reviewer 1 Report
Reviewer Recommendation and Comments for manuscript coatings-2287726 with the title: “Redox and Corrosion Behavior of Poly(3,4-Ethylenedioxythiophene) Film on Stainless Steel in NaCl Solution”, “Effect of redox switch, coupling and continuous polarization on the anti-corrosion properties of PEDOT film in NaCl solution” authors: V.M. Udowo, M. Yan, F. Liu, E. Han.
It is true that we can have different opinions about certain phenomena/processes. From a technical point of view I have no further questions/clarifications. I'm just pointing out that there are some typos that need to be corrected.
e.g.
Q1
8. Line 90│”potential scan rate of 0.10 mV/S.”. The potential scan rate of 0.1 mV/s is too low for cyclic voltammetry. (s should be used instead of S). It must be explained why such a low rate was used.
R1
The unit of scan rate is “10 mV/s or 0.10 V/s” not “0.10 mV/s”. The explanation for the slow scan rate has been added to the manuscript:
The slow potential scan rate was chosen to allow sufficient time for the products of the reduction reaction in the forward scan to participate in the backward reaction and generate clear anodic and cathodic reversible peaks.
Q2
“10 mV/s or 0.10 V/s” 10 mV/s = 0.010 V/s!? Any kind of mistake must be avoided in order not to induce confusion, of any kind.
The quality of the figures must be improved.
Author Response
Response to Reviewer(s)' Comments:
Reviewer: #1
Q1.“10 mV/s or 0.10 V/s” 10 mV/s = 0.010 V/s!? Any kind of mistake must be avoided in order not to induce confusion, of any kind.
A new scan rate of 50 mV/s has been adopted for measurement of the CV curve. The curve is presented in Figure 3.
Q2. The quality of the figures must be improved.
Corrected.
Reviewer 3 Report
OK.
Author Response
Reviewer: #3
Q1. English language and style are fine/minor spell check required.
We have spell checked the words in the manuscript with regard to this recommendation.